# Residual Relaxation for Multi-view Representation Learning

**Yifei Wang**[1]    **Zhengyang Geng**[2]    **Feng Jiang**[2]    **Chuming Li**[3]
**Yisen Wang**[2,4*]    **Jiansheng Yang**[1]    **Zhouchen Lin**[2,4,5]

[1] School of Mathematical Sciences, Peking University, China
[2] Key Lab. of Machine Perception, School of Artificial Intelligence, Peking University, Beijing, China
[3] School of Engineering, The University of Sydney, Australia
[4] Institute for Artificial Intelligence, Peking University, Beijing, China
[5] Pazhou Lab, Guangzhou, China

## Abstract

Multi-view methods learn representations by aligning multiple views of the same image and their performance largely depends on the choice of data augmentation. In this paper, we notice that some other useful augmentations, such as image rotation, are harmful for multi-view methods because they cause a semantic shift that is too large to be aligned well. This observation motivates us to relax the exact alignment objective to better cultivate stronger augmentations. Taking image rotation as a case study, we develop a generic approach, Pretext-aware Residual Relaxation (Prelax), that relaxes the exact alignment by allowing an adaptive residual vector between different views and encoding the semantic shift through pretext-aware learning. Extensive experiments on different backbones show that our method can not only improve multi-view methods with existing augmentations, but also benefit from stronger image augmentations like rotation.

## 1 Introduction

Without access to labels, self-supervised learning relies on surrogate objectives to extract meaningful representations from unlabeled data, and the chosen surrogate objectives largely determine the quality and property of the learned representations [24, 19]. Recently, multi-view methods have become a dominant approach for self-supervised representation learning that achieves impressive downstream performance, and many modern variants have been proposed [22, 14, 1, 23, 2, 12, 3, 4, 11, 5]. Nevertheless, most multi-view methods can be abstracted and summarized as the following pipeline: for each input $\mathbf{x}$, we apply several (typically two) random augmentations to it, and learn to align these different "views" $(\mathbf{x}_1, \mathbf{x}_2, \dots)$ of $\mathbf{x}$ by minimizing their distance in the representation space.

In multi-view methods, the pretext, *i.e.,* image augmentation, has a large effect on the final performance. Typical choices include image re-scaling, cropping, color jitters, *etc* [2]. However, we find that some augmentations, for example, image rotation, is seldom utilized in state-of-the-art multi-view methods. Among these augmentations, Figure 1a shows that rotation causes severe accuracy drop in a standard supervised model. Actually, image rotation is a stronger augmentation that largely affects the image semantics, and as a result, enforcing exact alignment of two different rotation angles could degrade the representation ability in existing multi-view methods. Nevertheless, it does not mean that strong augmentations cannot provide useful semantics for representation learning. In fact, rotation is known as an effective signal for predictive learning [10, 30, 21]. Differently, predictive methods learn representations by predicting the pretext (*e.g.,* rotation angle) from the cor-

---

*Corresponding author: Yisen Wang (yisen.wang@pku.edu.cn).

35th Conference on Neural Information Processing Systems (NeurIPS 2021).

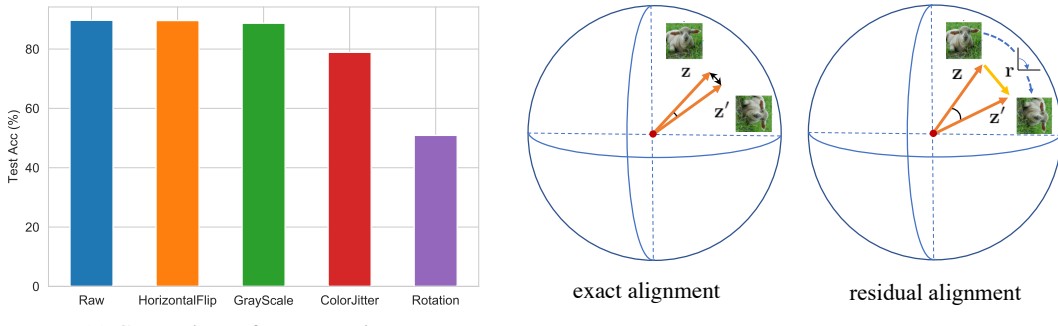

(a) Comparison of augmentations.

(b) A toy example of residual relaxation.

Figure 1: Left: the effect of different augmentations of CIFAR-10 test images with a supervised model (trained without using any data augmentation, more details in Appendix A). Right: an illustration of the exact alignment objective of multi-view methods ($\mathbf{z}' \rightarrow \leftarrow \mathbf{z}$) and the relaxed residual alignment of our Prelax ($\mathbf{z}' - \mathbf{r} \rightarrow \leftarrow \mathbf{z}$). As the rotation largely modifies the image semantics, our Prelax adopts a rotation-aware residual vector $\mathbf{r}$ to bridge the representation of two different views.

responding view. In this way, the model is encouraged to encode pretext-aware image semantics, which also yields good representations.

To summarize, strong augmentations like rotation carry meaningful semantics, while being harmful for existing multi-view methods due to large semantic shift. To address this dilemma, in this paper, we propose a generic approach that generalizes multi-view methods to cultivating stronger augmentations. Drawing inspirations from the soft-margin SVM, we propose *residual alignment*, which relaxes the exact alignment in multi-view methods by incorporating a residual vector between two views. Besides, we develop a predictive loss for the residual to ensure that it encodes the semantic shift between views (*e.g.,* image rotation). We name this technique as Pretext-aware REsidual ReLAXation (Prelax), and an illustration is shown in Figure 1b. Prelax serves as a generalized multi-view method that is adaptive to large semantic shift and combines image semantics extracted from both pretext-invariant and pretext-aware methods. We summarize our contributions as follows:

- We propose a generic technique, Pretext-aware Residual Relaxation (Prelax), that generalizes multi-view representation learning to benefit from stronger image augmentations.

- Prelax not only extracts pretext-invariant features as in multi-view methods, but also encodes pretext-aware features into the pretext-aware residuals. Thus, it can serve as a unified approach to bridge the two existing methodologies for representation learning.

- Experiments show that Prelax can bring significant improvement over both multi-view and predictive methods on a wide range of benchmark datasets.

## 2 Related Work

**Multi-view Learning.** Although multi-view learning could refer to a wider literature [16], here we restrict our discussions to the context of Self-Supervised Learning (SSL), where multi-view methods learn representations by aligning multiple views of the same image generated through random data augmentation [1]. There are two kinds of methods to keep the representations well separated: contrastive methods, which achieve this by maximizing the difference between different samples [2, 12], and similarity-based methods, which prevent representation collapse via implicit mechanisms like predictor and gradient stopping [11, 5]. Although having lots of modern variants, multi-view methods share the same methodology, that is to extract features that are *invariant* to the predefined augmentations, *i.e.,* pretext-invariant features [20].

**Predictive Learning.** Another thread of methods is to learn representations by predicting self-generated surrogate labels. Specifically, it applies a transformation (*e.g.,* image rotation) to the input image and requires the learner to predict properties of the transformation (*e.g.,* the rotation angle) from the transformed images. As a result, the extracted image representations are encouraged to become aware of the applied pretext (*e.g.,* image rotation). Thus, we also refer to them as *pretext-*

*aware methods*. The pretext tasks can be various, to name a few, Rotation [10], Jigsaw [21], Relative Path Location [8], Colorization [30].

**Generalized Multi-view Learning.** Although there are plenty of works on each branch, how to bridge the two methodologies remains under-explored. Prior to our work, there are only a few works on this direction. Some directly combine AMDIM (pretext-invariant) [1] and Rotation (pretext-aware) [10] objectives [9]. However, a direct combination of the two contradictory objectives may harm the final representation. LooC [28] proposes to separate the embedding space to several parts, where each subspace learns local invariance *w.r.t.* a specific augmentation. But this is achieved at the cost of limiting the representation flexibility of each pretext to the predefined subspace. Different from them, our proposed Prelax provides a more general solution by allowing an adaptive residual vector to encode the semantic shift. In this way, both kinds of features are encoded in the same representation space.

# 3 The Proposed Pretext-aware Residual Relaxation (Prelax) Method

## 3.1 Preliminary

**Problem Formulation.** Given unlabeled data $\{\mathbf{x}_i\}$, unsupervised representation learning aims to learn an encoder network $\mathcal{F}_{\boldsymbol{\theta}}$ that extracts meaningful low-dimensional representations $\mathbf{z} \in \mathbb{R}^{d_z}$ from high-dimensional input images $\mathbf{x} \in \mathbb{R}^{d_x}$. The learned representation is typically evaluated on a downstream classification task by learning a linear classifier with labeled data $\{\mathbf{x}_i, y_i\}$.

**Multi-view Representation Learning.** For an input image $\mathbf{x} \in \mathbb{R}^{d_x}$, we can generate a different view by data augmentation, $\mathbf{x}' = t(\mathbf{x})$, where $t \in \mathcal{T}$ refers to a randomly drawn augmentation operator from the pretext set $\mathcal{T}$. Then, the transformed input $\mathbf{x}'$ and the original input $\mathbf{x}$ are passed into an online network $\mathcal{F}_{\boldsymbol{\theta}}$ and a target network $\mathcal{F}_{\boldsymbol{\phi}}$, respectively. Optionally, the output of the online network is further processed by an MLP predictor network $\mathcal{G}_{\boldsymbol{\theta}}$, to match the output of the target network. As two different views of the same image (*i.e.,* positive samples), $\mathbf{x}$ and $\mathbf{x}'$ should have similar representations, so we align their representations with the following similarity loss,

$$\mathcal{L}_{\text{sim}}(\mathbf{x}', \mathbf{x}; \boldsymbol{\theta}) = \|\mathcal{G}_{\boldsymbol{\theta}}\left(\mathcal{F}_{\boldsymbol{\theta}}(\mathbf{x}')\right) - \mathcal{F}_{\boldsymbol{\phi}}(\mathbf{x})\|_2^2. \tag{1}$$

The representations, *e.g.,* $\mathbf{z} = \mathcal{F}_{\boldsymbol{\theta}}(\mathbf{x})$, are typically projected to a unit spherical ball before calculating the distance $(\mathbf{z}/\|\mathbf{z}\|_2)$, which makes the $\ell_2$ distance equivalent to the cosine similarity [2].

**Remark.** Aside from the similarity loss between positive samples, contrastive methods [22, 14, 23, 2] further encourage representation uniformity with an additional regularization minimizing the similarity between input and an independently drawn negative sample. Nevertheless, some recent works find that the similarity loss alone already suffices [11, 5]. In this paper, we mainly focus on improving the alignment between positive samples in the similarity loss. It can also be easily extended to contrastive methods by considering the dissimilarity regularization.

## 3.2 Objective Formulation

As we have noticed, the augmentation sometimes may bring a certain amount of semantic shift. Thus, enforcing exact alignment of different views may hurt the representation quality, particularly when the data augmentation is too strong for the positive pairs to be matched exactly. Therefore, we need to relax the exact alignment in Eq. (1) to account for the semantic shift brought by the data augmentation.

**Residual Relaxed Similarity Loss.** Although the representations may not align exactly, *i.e.,* $\mathbf{z}' \neq \mathbf{z}$, however, the *representation identity* will always hold: $\mathbf{z}' - (\mathbf{z}' - \mathbf{z}) = \mathbf{z}$, where $\mathbf{z}' - \mathbf{z}$ represents the shifted semantics by augmentation. This makes this identity a proper candidate for multi-view alignment under various augmentations as long as the shifted semantic is taken into consideration.

Specifically, we replace the exact alignment (denoted as $\rightarrow\leftarrow$) in the similarity loss (Eq. (1)) by the proposed *identity alignment*, *i.e.,*

$$\mathcal{G}_{\boldsymbol{\theta}}(\mathbf{z}'_{\boldsymbol{\theta}}) \rightarrow\leftarrow \mathbf{z}_{\boldsymbol{\phi}} \quad \Rightarrow \quad \mathcal{G}_{\boldsymbol{\theta}}(\mathbf{z}'_{\boldsymbol{\theta}}) - \mathcal{G}_{\boldsymbol{\theta}}(\mathbf{r}) \rightarrow\leftarrow \mathbf{z}_{\boldsymbol{\phi}}, \tag{2}$$

where we include a residual vector $\mathbf{r} \stackrel{\Delta}{=} \mathbf{z}'_{\boldsymbol{\theta}} - \mathbf{z}_{\boldsymbol{\theta}} = \mathcal{F}_{\boldsymbol{\theta}}(\mathbf{x}') - \mathcal{F}_{\boldsymbol{\theta}}(\mathbf{x})$ to represent the difference on the representations. To further enable a better tradeoff between the exact and identity alignments,

we have the following *residual alignment*:

$$\mathcal{G}_{\boldsymbol{\theta}}(\mathbf{z}'_{\boldsymbol{\theta}}) - \alpha\mathcal{G}_{\boldsymbol{\theta}}(\mathbf{r}) \rightarrow\leftarrow \mathbf{z}_{\boldsymbol{\phi}}, \tag{3}$$

where $\alpha \in [0,1]$ is the interpolation parameter. When $\alpha = 0$, we recover the exact alignment; when $\alpha = 1$, we recover the identity alignment. We name the corresponding learning objective as the Residual Relaxed Similarity (R2S) loss, which minimizes the squared $\ell_2$ distance among two sides:

$$\mathcal{L}^{\alpha}_{\mathrm{R2S}}(\mathbf{x}', \mathbf{x}; \boldsymbol{\theta}) = \|\mathcal{G}_{\boldsymbol{\theta}}(\mathcal{F}_{\boldsymbol{\theta}}(\mathbf{x}')) - \alpha\mathcal{G}_{\boldsymbol{\theta}}(\mathbf{r}) - \mathcal{F}_{\boldsymbol{\phi}}(\mathbf{x})\|_2^2. \tag{4}$$

**Predictive Learning (PL) Loss.** To ensure the relaxation works as expected, the residual $\mathbf{r}$ should encode the semantic shift caused by the augmentation, *i.e.,* the pretext. Inspired by predictive learning [10], we utilize the residual to predict the corresponding augmentation for its pretext-awareness. In practice, the assigned parameters for the random augmentation $\mathbf{t}$ can be generally divided into the discrete categorical variables $\mathbf{t}^d$ (*e.g.,* flipping or not, graying or not), and the continuous variables $\mathbf{t}^c$ (*e.g.,* scale, ratio, jittered brightness). Thus, we learn a PL predictor $\mathcal{H}_{\boldsymbol{\theta}}$ to predict $(\mathbf{t}^d, \mathbf{t}^c)$ with cross entropy loss (CE) and mean square error loss (MSE), respectively:

$$\mathcal{L}_{\mathrm{PL}}(\mathbf{x}', \mathbf{x}, \mathbf{t}; \boldsymbol{\theta}) = \mathrm{CE}(\mathcal{H}^d_{\boldsymbol{\theta}}(\mathbf{r}), \mathbf{t}^d) + \|\mathcal{H}^c_{\boldsymbol{\theta}}(\mathbf{r}) - \mathbf{t}^c\|_2^2. \tag{5}$$

**Constraint on the Similarity.** Different from the exact alignment, the residual vector can be unbounded, *i.e.,* the difference between views can be arbitrarily large. This is not reasonable as the two views indeed share many common semantics. Therefore, we should utilize this prior knowledge to prevent the bad cases under residual similarity and add the following constraint

$$\mathcal{L}_{\mathrm{sim}} = \|\mathcal{G}_{\boldsymbol{\theta}}(\mathcal{F}_{\boldsymbol{\theta}}(\mathbf{x}')) - \mathcal{F}_{\boldsymbol{\phi}}(\mathbf{x})\|_2^2 \leq \varepsilon, \tag{6}$$

where $\varepsilon$ denotes the maximal degree of mismatch allowed between positive samples.

**The Overall Objective of Prelax.** By combining the three components above, we can reliably encode the semantic shift between augmentations while ensuring a good alignment between views:

$$\min_{\boldsymbol{\theta}} \ \mathcal{L}^{\alpha}_{\mathrm{R2S}}(\mathbf{x}', \mathbf{x}; \boldsymbol{\theta}) + \gamma\mathcal{L}_{\mathrm{PL}}(\mathbf{x}', \mathbf{x}; \boldsymbol{\theta}),$$
$$s.t. \quad \|\mathcal{G}_{\boldsymbol{\theta}}(\mathcal{F}_{\boldsymbol{\theta}}(\mathbf{x}')) - \mathcal{F}_{\boldsymbol{\phi}}(\mathbf{x})\|_2^2 \leq \varepsilon. \tag{7}$$

For simplicity, we transform it into a Lagrangian objective with a fixed multiplier $\beta \geq 0$, and obtain the overall Pretext-aware REsidual ReLAXation (Prelax) objective,

$$\mathcal{L}^{\alpha}_{\mathrm{R2S}}(\mathbf{x}', \mathbf{x}; \boldsymbol{\theta}) + \gamma\mathcal{L}_{\mathrm{PL}}(\mathbf{x}', \mathbf{x}; \boldsymbol{\theta}) + \beta\mathcal{L}_{\mathrm{sim}}(\mathbf{x}', \mathbf{x}; \boldsymbol{\theta}), \tag{8}$$

where $\alpha$ tradeoffs between the exact and identity alignments, $\gamma$ adjusts the amount of pretext-awareness of the residual, and $\beta$ controls the degree of similarity between positive pairs. An illustrative diagram of the Prelax objective is shown in Figure 1.

**Discussions.** In fact, there are other alternatives to relax the exact alignment. For example, we can utilize a margin loss

$$\mathcal{L}_{\mathrm{margin}}(\mathbf{x}', \mathbf{x}; \boldsymbol{\theta}) = \max(\|\mathcal{G}_{\boldsymbol{\theta}}(\mathcal{F}_{\boldsymbol{\theta}}(\mathbf{x}')) - \mathcal{F}_{\boldsymbol{\phi}}(\mathbf{x})\|_2^2 - \eta, 0), \tag{9}$$

where $\eta > 0$ is a threshold for the mismatch tolerance. However, it has two main drawbacks: 1) as each image and augmentation have different semantics, it is hard to choose a universal threshold for all images; and 2) the representation keeps shifting along with the training progress, making it even harder to maintain a proper threshold dynamically. Thus, a good relaxation should be adaptive to the training progress and the aligning of different views. While our Prelax adopts *pretext-aware residual vector*, which is learnable, flexible, and semantically meaningful.

### 3.3 Theoretical Analysis

As Prelax encodes both pretext-invariant and pretext-aware features, it can be semantically richer than both multi-view learning and predictive learning. Following the information-theoretic framework developed by [25], we show that Prelax provably enjoys better downstream performance.

We denote the random variable of input as $\mathbf{X}$ and learn a representation $\mathbf{Z}$ through a deterministic encoder $\mathcal{F}_{\boldsymbol{\theta}}$: $\mathbf{Z} = \mathcal{F}_{\boldsymbol{\theta}}(\mathbf{X})^2$. The representation $\mathbf{Z}$ is evaluated for a downstream task $\mathbf{T}$ by learning

---

[2]We use capitals to denote the random variable, *e.g.,* $\mathbf{X}$, and use lower cases to denote its outcome, *e.g.,* $\mathbf{x}$.

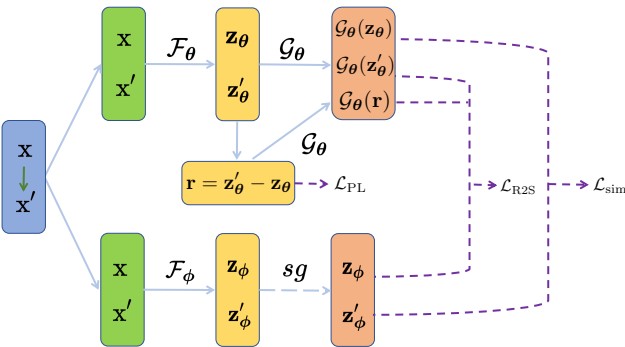

Figure 2: A diagram of our proposed Prelax objective. An image $\mathbf{x}$ is firstly augmented as $\mathbf{x}'$. Then the positive pair $(\mathbf{x}, \mathbf{x}')$, is processed by the online network $\mathcal{F}_\theta$ and the target network $\mathcal{F}_\phi$, respectively. Output of the online network is further processed by the target network $\mathcal{G}_\theta$, and the gradient of $\mathcal{F}_\phi$ is detached, *i.e.,* stop_grad, denoted as sg. Then the outputs are used to compute the three objectives, $\mathcal{L}_{\text{R2S}}$ (Eq. 4), $\mathcal{L}_{\text{PL}}$ (Eq. 5), and $\mathcal{L}_{\text{sim}}$ (Eq. 1) in the Prelax objective (Eq. 7).

a classifier on top of $\mathbf{Z}$. From an information-theoretic learning perspective, a desirable algorithm should maximize the Mutual Information (MI) between $\mathbf{Z}$ and $\mathbf{T}$, *i.e.,* $I(\mathbf{Z};\mathbf{T})$ [6]. Supervised learning on task $\mathbf{T}$ can learn representations by directly maximizing $I(\mathbf{Z};\mathbf{T})$. Without access to the labels $\mathbf{T}$, unsupervised learning resorts to maximizing $I(\mathbf{Z};\mathbf{S})$, where $\mathbf{S}$ denotes the surrogate signal $\mathbf{S}$ designed by each method. Specifically, multi-view learning matches $\mathbf{Z}$ with the randomly augmented view, denoted as $\mathbf{S}_v$; while predictive learning uses $\mathbf{Z}$ to predict the applied augmentation, denoted as $\mathbf{S}_a$. In Prelax, as we combine both semantics, we actually maximize the MI *w.r.t.* their joint distribution, *i.e.,* $I(\mathbf{Z};\mathbf{S}_v,\mathbf{S}_a)$. We denote the representations learned by supervised learning, multi-view learning, predictive learning, and Prelax as $\mathbf{Z}_{\text{sup}}, \mathbf{Z}_{\text{mv}}, \mathbf{Z}_{\text{PL}}, \mathbf{Z}_{\text{Prelax}}$, respectively.

**Theorem 1.** *Assume that by maximizing the mutual information, each method can retain all information in $\mathbf{X}$ about the learning signal $\mathbf{S}$ (or $\mathbf{T}$), i.e., $I(\mathbf{X};\mathbf{S}) = \max_{\mathbf{Z}} I(\mathbf{Z};\mathbf{S})$. Then we have the following inequalities on their task-relevant information $I(\mathbf{Z};\mathbf{T})$:*

$$I(\mathbf{X};\mathbf{T}) = I(\mathbf{Z}_{\text{sup}};\mathbf{T}) \geq I(\mathbf{Z}_{\text{Prelax}};\mathbf{T}) \geq \max(I(\mathbf{Z}_{\text{mv}};\mathbf{T}), I(\mathbf{Z}_{\text{PL}};\mathbf{T})). \tag{10}$$

**Theorem 2.** *Further assume that $\mathbf{T}$ is a $K$-class categorical variable. In general, we have the upper bound $u^e$ on the downstream Bayes errors $P^e := \mathbb{E}_{\mathbf{z}}\left[1 - \max_{\mathbf{t} \in \mathbf{T}} P\left(\mathbf{T} = \mathbf{t}|\mathbf{z}\right)\right]$,*

$$\bar{P}^e \leq u^e := \log 2 + P_{\text{sup}}^e \cdot \log K + I(\mathbf{X};\mathbf{T}|\mathbf{S}). \tag{11}$$

*where $\bar{P}^e = \text{Th}(P^e) = \min\{\max\{P^e, 0\}, 1 - 1/K\}$ denotes the thresholded Bayes error. Accordingly, we have the following inequalities on the upper bounds for different unsupervised methods,*

$$u_{\text{sup}}^e \leq u_{\text{Prelax}}^e \leq \min(u_{\text{mv}}^e, u_{\text{PL}}^e) \leq \max(u_{\text{mv}}^e, u_{\text{PL}}^e). \tag{12}$$

Theorem 1 shows that Prelax extracts more task-relevant information than multi-view and predictive methods, and Theorem 2 further shows that Prelax has a tighter upper bound on the downstream Bayes error. Therefore, Prelax is indeed theoretically superior to previous unsupervised methods by utilizing both pretext-invariant and pretext-aware features. Proofs are in Appendix B.

## 4  Practical Implementation

In this part, we present three practical variants of Prelax to generalize existing multi-view backbones: 1) one with existing multi-view augmentations (Prelax-std); 2) one with a stronger augmentation, image rotation (Prelax-rot); and 3) one with previous two strategies (Prelax-all).

### 4.1  Backbone

BYOL [11] and SimSiam [5] are both similarity-based methods and they differ mainly in the design of the target network $\mathcal{F}_\phi$. BYOL [11] utilizes momentum update of the target parameters $\phi$ from

the online parameters $\boldsymbol{\theta}$, *i.e.,* $\boldsymbol{\phi} \leftarrow \tau\boldsymbol{\phi} + (1 - \tau)\boldsymbol{\theta}$, where $\tau \in [0, 1]$ is the target decay rate. While SimSiam [5] simply regards the (stopped-gradient) online network as the target network, *i.e.,* $\boldsymbol{\phi} \leftarrow \mathrm{sg}(\boldsymbol{\theta})$. We mainly take SimSiam for discussion and our analysis also applies to BYOL.

For a given training image $\mathbf{x}$, SimSiam draws two random augmentations $(t_1, t_2)$ and get two views $(\mathbf{x}_1, \mathbf{x}_2)$, respectively. Then, SimSiam maximizes the similarity of their representations with a dual objective, where the two views can both serve as the input and the target to each other,

$$\mathcal{L}_{\mathrm{Simsiam}}(\mathbf{x}; \boldsymbol{\theta}) = \|\mathcal{G}_{\boldsymbol{\theta}}(\mathcal{F}_{\boldsymbol{\theta}}(\mathbf{x}_1)) - \mathcal{F}_{\boldsymbol{\phi}}(\mathbf{x}_2)\|_2^2 + \|\mathcal{G}_{\boldsymbol{\theta}}(\mathcal{F}_{\boldsymbol{\theta}}(\mathbf{x}_2)) - \mathcal{F}_{\boldsymbol{\phi}}(\mathbf{x}_1)\|_2^2. \quad (13)$$

## 4.2 Prelax-std

To begin with, we can directly generalize the baseline method with our Prelax method under existing multi-view augmentation strategies [2, 11]. For the same positive pair $(\mathbf{x}_1, \mathbf{x}_2)$, we can calculate their residual vector $\mathbf{r}_{12} = \mathcal{F}_{\boldsymbol{\theta}}(\mathbf{x}_1) - \mathcal{F}_{\boldsymbol{\theta}}(\mathbf{x}_2)$ and use it for the R2S loss (Eq. (4))

$$\mathcal{L}_{\mathrm{R2S}}^{\alpha}(\mathbf{x}_1, \mathbf{x}_2; \boldsymbol{\theta}) = \|\mathcal{G}_{\boldsymbol{\theta}}(\mathcal{F}_{\boldsymbol{\theta}}(\mathbf{x}_1)) - \alpha\mathcal{G}_{\boldsymbol{\theta}}(\mathbf{r}_{12}) - \mathcal{F}_{\boldsymbol{\phi}}(\mathbf{x}_2)\|_2^2. \quad (14)$$

We note that there is no difference in using $\mathbf{r}_{12}$ or $\mathbf{r}_{21}$ as the two views are dual. Then, we can adopt the similarity loss in the reverse direction as our similarity constraint loss,

$$\mathcal{L}_{\mathrm{sim}}(\mathbf{x}_2, \mathbf{x}_1; \boldsymbol{\theta}) = \|\mathcal{G}_{\boldsymbol{\theta}}(\mathcal{F}_{\boldsymbol{\theta}}(\mathbf{x}_2)) - \mathcal{F}_{\boldsymbol{\phi}}(\mathbf{x}_1)\|_2^2. \quad (15)$$

At last, we use the residual $\mathbf{r}_{12}$ for the PL loss to predict the augmentation parameters of $\mathbf{x}_1$, *i.e.,* $\mathbf{t}_1$, because $\mathbf{r}_{12} = \mathbf{z}_1 - \mathbf{z}_2$ directs towards $\mathbf{z}_1$. Combining the three losses above, we obtain our Prelax-std objective,

$$\mathcal{L}_{\mathrm{Prelax-std}}(\mathbf{x}; \boldsymbol{\theta}) = \mathcal{L}_{\mathrm{R2S}}^{\alpha}(\mathbf{x}_1, \mathbf{x}_2; \boldsymbol{\theta}) + \gamma\mathcal{L}_{\mathrm{PL}}(\mathbf{x}_1, \mathbf{x}_2, \mathbf{t}_1; \boldsymbol{\theta}) + \beta\mathcal{L}_{\mathrm{sim}}(\mathbf{x}_2, \mathbf{x}_1; \boldsymbol{\theta}). \quad (16)$$

## 4.3 Prelax-rot

As mentioned previously, with our residual relaxation we can benefit from stronger augmentations that are harmful for multi-view methods. Here, we focus on the image rotation example and propose the Prelax-rot objective with rotation-aware residual vector. To achieve this, we further generalize existing dual-view methods by incorporating a *third* rotation view.

Specifically, given two views $(\mathbf{x}_1, \mathbf{x}_2)$ generated with existing multi-view augmentations, we additionally draw a random rotation angle $a \in \mathcal{R} = \{0°, 90°, 180°, 270°\}$ and apply it to rotate $\mathbf{x}_1$ clockwise, leading to the third view $\mathbf{x}_3$. Note that the only difference between $\mathbf{x}_3$ and $\mathbf{x}_1$ is the rotation semantic $a$. Therefore, if we substitute $\mathbf{x}_1$ with $\mathbf{x}_3$ in the similarity loss, we should add a rotation-aware residual $\mathbf{r}_{31} = \mathbf{z}_3 - \mathbf{z}_1$ to bridge the gap. Motivated by this analysis, we propose the Rotation Residual Relaxation Similarity (R3S) loss,

$$\mathcal{L}_{\mathrm{R3S}}^{\alpha}(\mathbf{x}_{1:3}; \boldsymbol{\theta}) = \|\mathcal{G}_{\boldsymbol{\theta}}(\mathcal{F}_{\boldsymbol{\theta}}(\mathbf{x}_3)) - \alpha\mathcal{G}_{\boldsymbol{\theta}}(\mathbf{r}_{31}) - \mathcal{F}_{\boldsymbol{\phi}}(\mathbf{x}_2)\|_2^2. \quad (17)$$

which replace $\mathcal{G}_{\boldsymbol{\theta}}(\mathcal{F}_{\boldsymbol{\theta}}(\mathbf{x}_1))$ by its rotation-relaxed version $\mathcal{G}_{\boldsymbol{\theta}}(\mathcal{F}_{\boldsymbol{\theta}}(\mathbf{x}_3)) - \alpha\mathcal{G}_{\boldsymbol{\theta}}(\mathbf{r}_{31})$ in the similarity loss. Comparing the R2S loss (Eq. 14) and the R3S loss, we note that the relaxation of the R2S loss accounts for all the semantic shift between $\mathbf{x}_1$ and $\mathbf{x}_2$, while that of the R3S loss only accounts for the rotation augmentation between $\mathbf{x}_1$ and $\mathbf{x}_3$. Therefore, we could use the residual $\mathbf{r}_{31}$ to predict the rotation angle $a$ with the following RotPL loss for its rotation-awareness:

$$\mathcal{L}_{\mathrm{PL}}^{\mathrm{rot}}(\mathbf{x}_1, \mathbf{x}_3, \mathbf{a}; \boldsymbol{\theta}) = \mathrm{CE}(\mathcal{H}_{\boldsymbol{\theta}}(\mathbf{r}_{31}), a). \quad (18)$$

Combining with the similarity constraint, we obtain the Prelax-rot objective:

$$\mathcal{L}_{\mathrm{Prelax-rot}}(\mathbf{x}; \boldsymbol{\theta}) = \mathcal{L}_{\mathrm{R3S}}^{\alpha}(\mathbf{x}_{1:3}; \boldsymbol{\theta}) + \gamma\mathcal{L}_{\mathrm{PL}}^{\mathrm{rot}}(\mathbf{x}_1, \mathbf{x}_3, a; \boldsymbol{\theta}) + \beta\mathcal{L}_{\mathrm{sim}}(\mathbf{x}_2, \mathbf{x}_1; \boldsymbol{\theta}). \quad (19)$$

## 4.4 Prelax-all

We have developed Prelax-std that cultivates existing multi-view augmentations and Prelax-rot that incorporates image rotation. Here, we further utilize both existing augmentations and image rotation by combining the two objectives together, denoted as Prelax-all:

$$\begin{aligned}
\mathcal{L}_{\mathrm{Prelax-all}}(\mathbf{x}; \boldsymbol{\theta}) = &\frac{1}{2}\left(\mathcal{L}_{\mathrm{R2S}}^{\alpha_1}(\mathbf{x}_1, \mathbf{x}_2; \boldsymbol{\theta}) + \mathcal{L}_{\mathrm{R3S}}^{\alpha_2}(\mathbf{x}_{1:3}; \boldsymbol{\theta})\right) + \frac{\gamma_1}{2}\mathcal{L}_{\mathrm{PL}}(\mathbf{x}_1, \mathbf{x}_2, \mathbf{t}_1; \boldsymbol{\theta}) \\
&+ \frac{\gamma_2}{2}\mathcal{L}_{\mathrm{PL}}^{\mathrm{rot}}(\mathbf{x}_1, \mathbf{x}_3, a; \boldsymbol{\theta}) + \beta\mathcal{L}_{\mathrm{sim}}(\mathbf{x}_2, \mathbf{x}_1; \boldsymbol{\theta}),
\end{aligned} \quad (20)$$

where $\alpha_1, \alpha_2, \gamma_1, \gamma_2$ denotes the coefficients for R2S, R3S, PL and RotPL losses, respectively.

### 4.5 Discussions

Here we design three practical versions as different implementations of our generic framework of residual relaxation. Among them, Prelax-std focuses on further cultivating existing augmentation strategies, Prelax-rot is to incorporate the stronger (potentially harmful) rotation augmentation, while Prelax-all combines them all. Through the three versions, we demonstrate the wide applicability of Prelax as a generic framework. As for practical users, they could also adapt Prelax to their own application by incorporating specific domain knowledge. In this paper, as we focus on natural images, we take rotation as a motivating example as it is harmful for natural images. Nevertheless, rotation is not necessarily harmful in other domains, *e.g.,* medical images. Instead, random cropping could instead be very harmful for medical images as the important part could lie in the corner. In this scenario, our residual relaxation mechanism could also be used to encode the semantic shift caused by cropping and alleviate its bad effects.

## 5 Experiments

**Datasets.** Due to computational constraint, we carry out experiments on a range of medium-sized real-world image datasets, including well known benchmarks like CIFAR-10 [15], CIFAR-100 [15], and two ImageNet variants: Tiny-ImageNet-200 (200 classes with image size resized to $32\times32$) [27] and ImageNette (10 classes with image size $128\times128$)[3].

**Backbones.** As Prelax is designed to be a generic method for generalizing existing multi-view methods, we implement it on two different multi-view methods, SimSiam [5] and BYOL [11]. Specifically, we notice that SimSiam reported results on CIFAR-10, while the official code of BYOL included results on ImageNette. For a fair comparison, we evaluate SimSiam and its Prelax variant on CIFAR-10, and evaluate BYOL and its Prelax variant on ImageNette. In addition, we evaluate SimSiam and its Prelax variant on two additional datasets CIFAR-100 and Tiny-ImageNet-200, which are more challenging because they include a larger number of classes. For computational efficiency, we adopt the ResNet-18 [13] backbone (adopted in SimSiam [5] for CIFAR-10) to benchmark our experiments. For a comprehensive comparison, we also experiment with larger backbones, like ResNet-34 [13], and the results are included in Appendix C.

**Setup.** For Prelax-std, we use the same data augmentations as SimSiam [2, 5] (or BYOL [11]), including RandomResizedCrop, RandomHorizontalFlip, ColorJitter, and RandomGrayscale, *etc* using the PyTorch notations. For Prelax-rot and Prelax-all, we further apply a random image rotation at last of the transformation, where the angles are drawn randomly from $\{0°, 90°, 180°, 270°\}$. To generate targets for the PL objective in Prelax, for each image, we collect the assigned parameters in each random augmentation, such as crop centers, aspect ratios, rotation angles, *etc*. More details can be found in Appendix A.

**Training.** For SimSiam and its Prelax variants, we follow the same hyperparameters in [5] on CIFAR-10. Specifically, we use ResNet-18 as the backbone network, followed by a 3-layer projection MLP, whose hidden and output dimension are both 2048. The predictor is a 2-layer MLP whose hidden layer and output dimension are 512 and 2048 respectively. We use SGD for pre-training with batch size 512, learning rate 0.03, momentum 0.9, weight decay $5 \times 10^{-4}$, and cosine decay schedule [18] for 800 epochs. For BYOL and its Prelax variants, we also adopt the ResNet-18 backbone, and the projector and predictor are 2-layer MLPs whose hidden layer and output dimension are 256 and 4096 respectively. Following the default hyper-parameters on ImageNette[4], we use LARS optimizer [29] to train 1000 epochs with batch size 256, learning rate 2.0, weight decay $1 \times 10^{-6}$ while excluding the biases and batch normalization parameters from both LARS adaptation and weight decay. For the target network, the exponential moving average parameter $\tau$ starts from $\tau_{\text{base}} = 0.996$ and increases to 1 during training. As for the Prelax objective, we notice that sometimes, adopting a reverse residual $\mathbf{r}_{21}$ in the R2S loss (Eq. (14)) can bring slightly better results, which could be due to the swapped prediction in SimSiam's dual objective (Eq. (13)). Besides, a naïve choice of Prelax coefficients already works well: $\alpha = 1, \beta = 1, \gamma = 0.1$ for Prelax-std and Prelax-rot, and $\alpha_1 = \alpha_2 = 1, \beta = 1, \gamma_1 = \gamma_2 = 0.1$ for Prelax-all. More discussion about the hyper-parameters of Prelax can be found in Appendix E.

---

[3] https://github.com/fastai/imagenette
[4] https://github.com/deepmind/deepmind-research/tree/master/byol

Table 1: Linear evaluation on CIFAR-10 (a) and ImageNette (b) with ResNet-18 backbone. TTA: Test-Time Augmentation.

(a) CIFAR-10.

| Method | Acc. (%) |
|---|---|
| Supervised [13] (re-produced) | 95.0 |
| Rotation [10] (re-produced) | 88.3 |
| BYOL [11] (re-produced) | 91.1 |
| SimCLR [2] | 91.1 |
| SimSiam [5] | 91.8 |
| **SimSiam + Prelax** | **93.4** |

(b) ImageNette.

| Method | Acc. (%) |
|---|---|
| Supervised | 91.0 |
| Supervised + TTA | 92.2 |
| BYOL [11] (ResNet-18) | 91.9 |
| BYOL [11] (ResNet-50) | 92.3 |
| **BYOL + Prelax (ResNet-18)** | **92.6** |

**Evaluation.** After unsupervised training, we evaluate the backbone network by fine-tuning a linear classifier on top of its representation with other model weights held fixed. For SimSiam and its Prelax variants, the linear classifier is trained on labeled data from scratch using SGD with batch size 256, learning rate 30.0, momentum 0.9 for 100 epochs. The learning rate decays by 0.1 at the 60-th and 80-th epochs. For BYOL and its Prelax variants, we use SGD with Nesterov momentum over 80 epochs using batch size 25, learning rate 0.5 and momentum 0.9. Besides the in-domain linear evaluation, we also evaluate its transfer learning performance on the out-of-domain data by learning a linear classifier on the labeled target domain data.

## 5.1 Performance on Benchmark Datasets

**CIFAR-10.** In Table 1a, we compare Prelax against previous multi-view methods (SimCLR [2], SimSiam [5], and BYOL [11]) and predictive methods (Rotation [10]) on CIFAR-10. We can see that multi-view methods are indeed better than predictive ones. Nevertheless, predictive learning alone (*e.g.,* Rotation) achieves quite good performance, indicating that pretext-aware features are also very useful. By encoding both pretext-invariant and pretext-aware features, Prelax outperforms previous methods by a large margin, and achieve state-of-the-art performance on CIFAR-10. A comparison of the learning dynamics between SimSiam and Prelax can be found in Appendix F.

**ImageNette.** Beside the SimSiam backbone, we further apply our Prelax loss to the BYOL framework [11] and evaluate the two methods on the ImageNette dataset. In Table 1b, Prelax also shows a clear advantage over BYOL. Specifically, it improves the ResNet-18 version of BYOL by 0.7%, and even outperforms the ResNet-50 version by 0.3%.

Here, we can see that Prelax yields significant improvement on two different datasets with two different backbone methods. Thus, Prelax could serve as a generic method for improving existing multi-view methods by encoding pretext-aware features into the residual relaxation. For completeness, we also evaluate Prelax on the large scale dataset, ImageNet [7], as well as its transferability to other kinds of downstream tasks, such as object detection and instance segmentation on MS COCO [17]. As shown in Appendix D, Prelax still consistently outperforms the baselines across all tasks.

## 5.2 Effectiveness of Prelax Variants

For a comprehensive comparison of the three variants of Prelax objectives (Prelax-std, Prelax-rot, and Prelax-all), we conduct controlled experiments on a range of datasets based on the SimSiam backbone. Except CIFAR-10, we also conduct experiments on CIFAR-100 and Tiny-ImageNet-200, which are more challenging with more classes of images. For a fair comparison, we use the same training and evaluation protocols across all tasks.

**In-domain Linear Evaluation.** As shown in Table 2a, our Prelax objectives outperform the multi-view objective consistently on all three datasets, where Prelax-all improves SimSiam by 1.6% on CIFAR-10, 3.1% on CIFAR-100, and 1.5% on Tiny-ImageNet-200. Besides, Prelax-std and Prelax-rot are also better than SimSiam in most cases. Thus, the pretext-aware residuals in Prelax indeed help encode more useful semantics.

Table 2: A detailed comparison of SimSiam [5] and Prelax (ours) across three datasets: CIFAR-10 (C10), CIFAR-100 (C100), and Tiny-ImageNet-200 (Tiny200) with the same hyper-parameters.

(a) In-domain linear evaluation.

| Method | CIFAR-10 | CIFAR-100 | Tiny-ImageNet-200 |
|---|---|---|---|
| SimSiam [5] | 91.8 | 66.9 | 47.7 |
| SimSiam + Prelax-std | 92.5 | 67.5 | 47.9 |
| SimSiam + Prelax-rot | 92.4 | 67.3 | 47.1 |
| SimSiam + Prelax-all | **93.4** | **70.0** | **49.2** |

(b) Out-of-domain linear evaluation.

| Method | C100 $\rightarrow$ C10 | Tiny200 $\rightarrow$ C10 | Tiny200 $\rightarrow$ C100 |
|---|---|---|---|
| SimSiam [5] | 44.1 | 43.9 | 21.8 |
| SimSiam + Prelax-std | **45.0** | **45.1** | 21.8 |
| SimSiam + Prelax-rot | **45.0** | **45.1** | 22.0 |
| SimSiam + Prelax-all | 44.9 | 44.6 | **22.1** |

Table 3: Linear evaluation results of possible mechanisms for generalized multi-view learning on CIFAR-10 with SimSiam backbone.

(a) Comparison against alternative options.

| Method | Acc. (%) |
|---|---|
| SimSiam [5] | 91.8 |
| SimSiam + margin loss | 91.9 |
| Rotation [10] | 88.3 |
| SimSiam + rotation aug. | 87.9 |
| SimSiam + Rotation loss | 91.7 |
| SimSiam + Prelax (ours) | **93.4** |

(b) Ablation study.

| Method | Acc. (%) |
|---|---|
| Sim (*i.e.,* SimSiam [5]) | 91.8 |
| Sim + PL | 92.2 |
| Sim + R2S | 91.5 |
| R2S + PL | 91.7 |
| Sim + PL + R2S (Prelax-std) | **92.5** |
| Sim + RotPL | 91.1 |
| Sim + R3S | 91.9 |
| R3S + RotPL | 79.8 |
| Sim + RotPL + R3S (Prelax-rot) | **92.4** |

**Out-of-domain Linear Evaluation.** Besides the in-domain linear evaluation, we also transfer the representations to a target domain. In the out-of-domain linear evaluation results shown in Table 2b, the Prelax objectives still have a clear advantage over the multi-view objective (SimSiam), while sometimes Prelax-std and Prelax-rot enjoy better transferred accuracy than Prelax-all.

### 5.3 Empirical Understandings of Prelax

**Comparison against Alternative Options.** In Table 3a, we compare Prelax against several other relaxation options. "SimSiam + margin" refers to the margin loss discussed in Eq. (9), which uses a scalar $\eta$ to relax the exact alignment in multi-view methods. Here we tune the margin $\eta = 0.5$ with the best performance. Nevertheless, it has no clean advantage over SimSiam. Then, we try several options for incorporating a strong augmentation and image rotation: 1) Rotation is the PL baseline by predicting rotation angles [10], which is inferior to multi-view methods (SimSiam). 2) "SimSiam + rotation aug." directly applies a random rotation augmentation to each view, and learn with the SimSiam loss. However, it leads to lower accuracy, showing that the image rotation, as a strong augmentation, will *hurt* the performance of multi-view methods. 3) "SimSiam + Rotation" directly combines the SimSiam loss and the Rotation loss for training, which is still ineffective. 4) Our Prelax shows a significant improvement over SimSiam and other variants, showing that the residual alignment is an effective mechanism for utilizing strong augmentations like rotation.

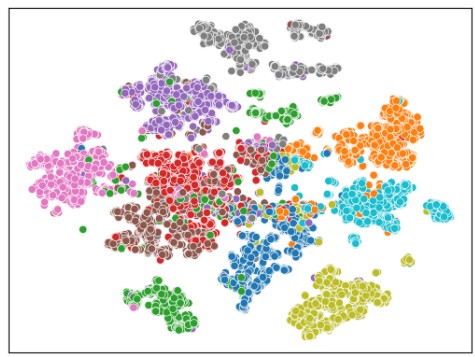
(a) Representation visualization.

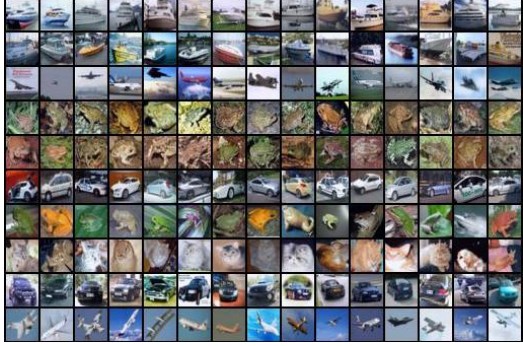
(b) Nearest image retrieval.

Figure 3: (a) Representation visualization of our Prelax on CIFAR-10 test set. Each point represents an image representation and its color denotes the class of the image. (b) On CIFAR-10 test set, given 10 random queries (not cherry-picked), we retrieve 15 nearest images in the representation space with Prelax (ours).

**Ablation Study.** We perform ablation study of each component of the Prelax objectives on CIFAR-10. From Table 3b, we notice that simply adding the PL loss alone cannot improve over SimSiam consistently, for example, Sim + RotPL causes 0.7 point drop in test accuracy. While with the help of our residual relaxation, we can improve over the baselines significantly and consistently, for example, Prelax-rot (Sim + RotPL + R3S) brings 0.6 point improvement on test accuracy. Besides, we can see that the PL loss is necessary by making the residual pretext-aware, without which the performance drops a lot, and the similarity constraint (Sim loss) is also important by avoiding bad cases when augmented images drift far from the anchor image. Therefore, the ablation study shows the residual relaxation loss, similarity loss, and PL loss all matter in our Prelax objectives.

### 5.4 Qualitative Analysis

**Representation Visualization.** To provide an intuitive understanding of the learned representations, we visualize them with t-SNE [26] on Figure 3a. We can see that in general, our Prelax can learn well-separated clusters of representations corresponding to the ground-truth image classes.

**Image Retrieval.** In Figure 3b, we evaluate Prelax on an image retrieval task. Given a random query image (not cherry-picked), the top-15 most similar images in representation space are retrieved, and the query itself is shown in the first column. We can see that although the unsupervised training with Prelax has no access to labels, the retrieved nearest images of Prelax are all correctly from the same class and semantically consistent with the query.

## 6 Conclusion

In this paper, we proposed a generic method, Prelax (Pretext-aware Residual Relaxation), to account for the (possibly large) semantic shift caused by image augmentations. With pretext-aware learning of the residual relaxation, our method generalizes existing multi-view learning by encoding both pretext-aware and pretext-invariant representations. Experiments show that our Prelax has outperformed existing multi-view methods significantly on a variety of benchmark datasets.

## Acknowledgement

Yisen Wang is partially supported by the National Natural Science Foundation of China under Grant 62006153, and Project 2020BD006 supported by PKU-Baidu Fund. Jiansheng Yang is supported by the National Science Foundation of China under Grant No. 11961141007. Zhouchen Lin was supported by the NSF China (No.s 61625301 and 61731018), NSFC Tianyuan Fund for Mathematics (No. 12026606) and Project 2020BD006 supported by PKU-Baidu Fund.

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
