# A  Experimental Details

**Evaluating Augmentations.** In Table 1, we compare different augmentations with a supervised ResNet-18 [7] on CIFAR-10 test set. Specifically, we first train a state-of-the-art supervised ResNet-18 with 95.01% test accuracy on CIFAR-10. [1] The supervised training uses no data augmentations. Afterwards, we evaluate the effect of different augmentations to the supervised model by applying each one (separately) to pre-process the test images of CIFAR-10. All of the included augmentations (except Rotation) belong to the augmentations used in SimSiam. For a fair comparison, we adopt the same configuration as in SimSiam and refer to the paper for more details. For Rotation, we adopt the same configuration as [4], where we sample a random rotation angle $\{0°, 90°, 180°, 270°\}$ and use it to rotate the raw image clock-wise.

**Data Augmentations and PL Targets.** We offer details of the augmentations by taking the SimSiam [1] variant of Prelax as an example. The BYOL [5] variants are implemented in the same way. For a fair comparison, we utilize the same augmentations in SimSiam [1], while collecting the augmentation parameters as the target variables for our Predictive Learning (PL) objective in Prelax. We adopt the PyTorch notations for simplicity. Specifically, for RandomResizedCrop, the operation randomly draws an $(i, j, h, k)$ pair, where $(i, j)$ denotes the center coordinates of the cropped region, while $(h, k)$ denotes the height and width of the cropped region. Accordingly, we calculate the relative coordinates, the area ratio, and the aspect ratio (relative to the raw image), as four continuous target variables. Similarly, the ColorJitter opration randomly samples four factors corresponding to the adjustment for brightness, contrast, saturation, hue, respectively. We collect them as four additional continuous target variables. As for operations like RandomHorizontalFlip, RandomGrayscale, RandomApply, they draw a binary variable with $0/1$ outcome according to a predefined probability $p$, and apply the augmentations if it is 1 and do nothing otherwise. We collect these random outcomes $(0/1)$ as discrete target variables. As for the rotation operation, we take the rotation angles randomly drawn from the set $\{0°, 90°, 180°, 270°\}$, as a discrete 4-class categorical variable.

# B  Theoretical Results and Proofs

Here, we provide the complete proof of the theoretical results in Section 3.3. More rigorously, we give the definition of minimal and sufficient representations for self-supervision [11], and give a more formal description of our results.

**Definition 1** (Minimal and Sufficient Representations for Signal **S**). *Let $\mathbf{Z}^*$ be the minimal and sufficient representation for self-supervised signal* **S** *if it satisfies the following conditions in the meantime: 1)* $\mathbf{Z}^*$ *is sufficient,* $\mathbf{Z}^* = \arg\max_{\mathbf{Z}} I(Z; S)$*; 2)* $\mathbf{Z}^*$ *is minimal, i.e.,* $\mathbf{Z}^* = \operatorname{argmin}_{\mathbf{Z}} H(\mathbf{Z}|\mathbf{S})$.

The following lemma shows that the maximal mutual information of $I(\mathbf{Z}^*, \mathbf{S})$ is $I(\mathbf{X}, \mathbf{S})$.

**Lemma 1.** *For a minimal and sufficient representation* **Z** *that is obtained with a deterministic encoder* $\mathcal{F}_{\boldsymbol{\theta}}$ *of input* **X** *with enough capacity, we have* $I(\mathbf{Z}^*; \mathbf{S}) = I(\mathbf{X}; \mathbf{S})$.

*Proof.* As the encoder $\mathcal{F}_{\boldsymbol{\theta}}$ is deterministic, it induces the following conditional independence: $\mathbf{S} \perp\!\!\!\perp \mathbf{Z} \mid \mathbf{X}$, which leads to the data processing Markov chain $\mathbf{S} \leftrightarrow \mathbf{X} \to \mathbf{Z}$. Accordingly to the data processing inequality (DIP) [2], we have $I(\mathbf{Z}; \mathbf{S}) \leq I(\mathbf{X}; \mathbf{S})$, and with enough model capacity in $\mathcal{F}_{\boldsymbol{\theta}}$, the sufficient and minimal representation $\mathbf{Z}^*$ will have $I(\mathbf{Z}^*; \mathbf{S}) = \max_{\mathbf{Z}} I(\mathbf{Z}; \mathbf{S}) = I(\mathbf{X}; \mathbf{S})$. □

In the main text, we introduce several kinds of learning signals, the target variable **T**, the multi-view signal $\mathbf{S}_v$, the predictive learning signal $\mathbf{S}_a$, and the joint signal $(\mathbf{S}_v, \mathbf{S}_a)$ used by our Prelax method. For clarity, we denote the learned *minimal and sufficient* representations as $\mathbf{Z}_{\text{sup}}$, $\mathbf{Z}_{\text{mv}}$, $\mathbf{Z}_{\text{PL}}$, $\mathbf{Z}_{\text{Prelax}}$, respectively.

Next, we restate Theorem 1 with the definitions above and provide a complete proof.

**Theorem 2** (restated). *We have the following inequalities on the four minimal and sufficient representations,* $\mathbf{Z}_{\text{sup}}$*,* $\mathbf{Z}_{\text{mv}}$*,* $\mathbf{Z}_{\text{PL}}$*,* $\mathbf{Z}_{\text{Prelax}}$ :

$$I(\mathbf{X}; \mathbf{T}) = I(\mathbf{Z}_{\text{sup}}; \mathbf{T}) \geq I(\mathbf{Z}_{\text{Prelax}}; \mathbf{T}) \geq \max(I(\mathbf{Z}_{\text{mv}}; \mathbf{T}), I(\mathbf{Z}_{\text{PL}}; \mathbf{T})). \tag{1}$$

---

[1] https://github.com/kuangliu/pytorch-cifar

*Proof.* By Lemma 1, we have the following properties in the self-supervised representations:

$$I(\mathbf{Z}_{\mathrm{mv}}; \mathbf{S}_v) = I(\mathbf{X}; \mathbf{S}_v), \ I(\mathbf{Z}_{\mathrm{PL}}; \mathbf{S}_a) = I(\mathbf{X}; \mathbf{S}_a), \ I(\mathbf{Z}_{\mathrm{Prelax}}; \mathbf{S}_v, \mathbf{S}_a) = I(\mathbf{X}; \mathbf{S}_v, \mathbf{S}_a). \quad (2)$$

Thus, for each minimal and sufficient self-supervised representation $\mathbf{Z} \in \{\mathbf{Z}_{\mathrm{mv}}, \mathbf{Z}_{\mathrm{PL}}, \mathbf{Z}_{\mathrm{Prelax}}\}$ and the corresponding signal $\mathbf{S} \in \{\mathbf{S}_v, \mathbf{S}_a, (\mathbf{S}_v, \mathbf{S}_a)\}$, we have,

$$I(\mathbf{Z}; \mathbf{S}; \mathbf{T}) = I(\mathbf{X}; \mathbf{S}; \mathbf{T}), \ I(\mathbf{Z}; \mathbf{S}|\mathbf{T}) = I(\mathbf{X}; \mathbf{S}|\mathbf{T}). \quad (3)$$

Besides, because $\mathbf{Z}$ is minimal, we also have,

$$I(\mathbf{Z}; \mathbf{T}|\mathbf{S}) \leq H(\mathbf{Z}|\mathbf{S}) = 0. \quad (4)$$

Together with the two equalities above, we further have the following equality on $I(\mathbf{Z}; \mathbf{T})$:

$$
\begin{aligned}
I(\mathbf{Z}; \mathbf{T}) &= I(\mathbf{Z}; \mathbf{T}; \mathbf{S}) + I(\mathbf{Z}; \mathbf{T}|\mathbf{S}) \\
&= I(\mathbf{X}; \mathbf{T}; \mathbf{S}) + \underbrace{I(\mathbf{Z}; \mathbf{T}|\mathbf{S})}_{0} \\
&= I(\mathbf{X}; \mathbf{T}) - I(\mathbf{X}; \mathbf{T}|\mathbf{S}) \\
&= I(\mathbf{Z}_{\mathrm{sup}}; \mathbf{T}) - I(\mathbf{X}; \mathbf{T}|\mathbf{S}).
\end{aligned} \quad (5)
$$

Therefore, the gap between supervised representation $\mathbf{Z}_{\mathrm{sup}}$ and each self-supervised representation $\mathbf{Z} \in \{\mathbf{Z}_{\mathrm{mv}}, \mathbf{Z}_{\mathrm{PL}}, \mathbf{Z}_{\mathrm{Prelax}}\}$ is $I(\mathbf{X}; \mathbf{T}|\mathbf{S})$, for which we have the following inequalities:

$$\max(I(\mathbf{X}; \mathbf{T}|\mathbf{S}_v), I(\mathbf{X}; \mathbf{T}|\mathbf{S}_a)) \geq \min(I(\mathbf{X}; \mathbf{T}|\mathbf{S}_v), I(\mathbf{X}; \mathbf{T}|\mathbf{S}_a)) \geq I(\mathbf{X}; \mathbf{T}|\mathbf{S}_v, \mathbf{S}_a). \quad (6)$$

Further combining with Lemma 1 and Eq. (5), we arrive at the inequalities on the target mutual information:

$$I(\mathbf{X}; \mathbf{T}) = I(\mathbf{Z}_{\mathrm{sup}}; \mathbf{T}) \geq I(\mathbf{Z}_{\mathrm{Prelax}}; \mathbf{T}) \geq \max(I(\mathbf{Z}_{\mathrm{mv}}; \mathbf{T}), I(\mathbf{Z}_{\mathrm{PL}}; \mathbf{T})), \quad (7)$$

which completes the proof. $\qquad\square$

**Remark.** Theorem 2 shows that the downstream performance gap between supervised representation $\mathbf{Z}_{\mathrm{sup}}$ and self-supervised representation $\mathbf{Z}$ is $I(\mathbf{X}; \mathbf{T}|\mathbf{S})$, *i.e.,* the information left in $\mathbf{X}$ about the target variable $\mathbf{T}$ except that in $\mathbf{S}$. Thus, if we choose a self-supervised signal $\mathbf{S}$ such that the left information is relatively small, we can guarantee a good downstream performance. Comparing the three self-supervised methods with learning signal $\mathbf{S}_v$, $\mathbf{S}_a$, and $(\mathbf{S}_v, \mathbf{S}_a)$, we can see that our Prelax utilizes more information in $\mathbf{X}$, and consequently, the left information $I(\mathbf{X}; \mathbf{T}|\mathbf{S}_v, \mathbf{S}_a)$ is smaller than both multi-view methods $I(\mathbf{X}; \mathbf{T}|\mathbf{S}_a)$ and predictive methods $I(\mathbf{X}; \mathbf{T}|\mathbf{S}_a)$.

In the following theorem, we further show that our Prelax has a tighter upper bound on the Bayes error of downstream classification tasks. To begin with, we prove a relationship between the supervised and self-supervised Bayes errors following [11].

**Lemma 3.** *Assume that $\mathbf{T}$ is a $K$-class categorical variable. We define the Bayes error on downstream task $T$ as*

$$P^e := \mathbb{E}_{\mathbf{z}}\left[1 - \max_{\mathbf{t} \in \mathbf{T}} P(\mathbf{T} = \mathbf{t}|\mathbf{z})\right]. \quad (8)$$

*Denote the Bayes error of self-supervised learning (SSL) methods with signal $\mathbf{S}$ as $P_{\mathrm{ssl}}^e$ and that of supervised methods as $P_{\mathrm{sup}}^e$. Then, we can show that the SSL Bayes error $P_{\mathrm{ssl}}^e$ can be upper bounded by the supervised Bayes error $P_{\mathrm{sup}}^e$, i.e.,*

$$\bar{P}_{\mathrm{ssl}}^e \leq u^e := \log 2 + P_{\mathrm{sup}}^e \cdot \log K + I(\mathbf{X}; \mathbf{T}|\mathbf{S}). \quad (9)$$

*where $\bar{P}^e = \mathrm{Th}(P^e) = \min\{\max\{P^e, 0\}, 1 - 1/K\}$ denotes the thresholded Bayes error in the feasible region, and $u^e$ denote the value of the upper bound.*

*Proof.* Denote the minimal and sufficient representations learned by SSL and supervised methods as $\mathbf{Z}_{\mathrm{ssl}}$ and $\mathbf{Z}_{\mathrm{sup}}$, respectively. We use two following inequalities from [3] and [2],

$$P_{\mathrm{ssl}}^e \leq -\log(1 - P_{\mathrm{ssl}}^e) \leq H(\mathbf{T} \mid \mathbf{Z}_{\mathrm{ssl}}), \quad (10)$$
$$H(\mathbf{T}|\mathbf{Z}_{\mathrm{sup}}) \leq \log 2 + P_{\mathrm{sup}}^e \log K. \quad (11)$$

Table 4: Linear evaluation accuracy (%) with ResNet-34 backbone.

| Method | CIFAR-10 | CIFAR-100 | Tiny-ImageNet-200 |
|---|---|---|---|
| SimSiam [1] | 91.2 | 60.9 | 39.0 |
| SimSiam + Prelax-std | 92.4 | 67.6 | 48.4 |
| SimSiam + Prelax-rot | 93.0 | 67.0 | 40.9 |
| SimSiam + Prelax-all | **93.9** | **69.3** | **49.4** |

Comparing $H(\mathbf{T}|\mathbf{Z})$ and $H(\mathbf{T}|\mathbf{Z}_{\text{sup}})$, together with Eq. (5), we can show that they are tied with the following equality,

$$
\begin{aligned}
H(\mathbf{T}|\mathbf{Z}_{\text{ssl}}) &= H(\mathbf{T}) - I(\mathbf{Z}_{\text{ssl}}; \mathbf{T}) \\
&= H(\mathbf{T}) - I(\mathbf{Z}_{\text{sup}}; \mathbf{T}) + I(\mathbf{X}; \mathbf{T}|\mathbf{S}) \\
&= H(\mathbf{T}|\mathbf{Z}_{\text{sup}}) + I(\mathbf{X}; \mathbf{T}|\mathbf{S}).
\end{aligned}
\tag{12}
$$

Further combining Eq. (10) & (11), we have

$$
\begin{aligned}
P_{\text{ssl}}^e &\le H\left(\mathbf{T} \mid \mathbf{Z}_{\text{ssl}}\right) \\
&= H(\mathbf{T}|\mathbf{Z}_{\text{sup}}) + I(\mathbf{X}; \mathbf{T}|\mathbf{S}) \\
&\le \log 2 + P_{\text{sup}}^e \log K + I(\mathbf{X}; \mathbf{T}|\mathbf{S}) := u^e,
\end{aligned}
\tag{13}
$$

which completes the proof. $\square$

Given the upper bound in Lemma 3, and the inequalities on the downstream performance gap $I(\mathbf{X}; \mathbf{T}|\mathbf{S})$ in Eq. (6), we will arrive at the following inequalities on the upper bounds on the self-supervised representations.

**Theorem 4** (restated). *We denote the the upper bound on the Bayes error of each representation, $\mathbf{Z}_{\text{sup}}, \mathbf{Z}_{\text{mv}}, \mathbf{Z}_{\text{PL}}, \mathbf{Z}_{\text{Prelax}}$, by $u_{\text{sup}}^e, u_{\text{mv}}^e, u_{\text{PL}}^e, u_{\text{Prelax}}^e$, respectively. Then, they satisfy the following inequalities:*

$$
u_{\text{sup}}^e \le u_{\text{Prelax}}^e \le \min(u_{\text{mv}}^e, u_{\text{PL}}^e) \le \max(u_{\text{mv}}^e, u_{\text{PL}}^e).
\tag{14}
$$

Theorem 4 shows that our Prelax enjoys a tighter lower bounds on downstream Bayes error than both multi-view methods and predictive methods.

## C Evaluation with Larger Backbone Networks

In the main text, we conduct experiments with the ResNet-18 backbone network. Here, for completeness, we further evaluate our Prelax with larger backbone networks. Specifically, for SimSiam variants, we evaluate the ResNet-34 [7] across three datasets, CIFAR-10, CIFAR-100, and Tiny-ImageNet-200. For a fair comparison, we adopt the same hyper-parameters as for the ResNet-18 backbone. As can be seen for Table 4, all our Prelax variants achieves better results than the Sim-Siam baseline on all three datasets. Specifically, we can see that our Prelax-all variant attains the best results and it achieves better results with a larger backbone. Besides, we also experiment with ResNet-50 for the BYOL variant, where our Prelax variant also achieves better performance by improving from 92.3% to 92.7%.

## D Evaluation on Large Scale Datasets

**Setup.** Although we cannot carry out the full ImageNet experiments with limited time and computation, we gather some preliminary results on the downsampled ImageNet dataset (128x128) with the ResNet-18 backbone. For a fair comparison, our experiments are conducted with the official code of BYOL. All models are trained for 100 epochs with the default hyperparameters.

**Evaluation protocol.** For downstream evaluation, we report both the linear evaluation task on ImageNet and two transfer learning tasks on the MS COCO dataset [10]. Specifically, we perform object detection on the standard RetinaNet [9] with FPN [8], and conduct instance segmentation on

Table 5: Evaluation of different pretraining methods on the downsampled ImageNet dataset (128x128) with ResNet-18 backbone.

| (a) Linear Evaluation. | | (b) Object Detection. | | | | (c) Instance Segmentation. | |
|---|---|---|---|---|---|---|---|
| Method | Acc (%) | Method | $AP_{50}$ | AP | $AP_{75}$ | Method | MAP |
| | | RandInit | 32.7 | 19.5 | 20.1 | RandInit | 15.8 |
| BYOL | 49.2 | BYOL | 36.6 | 22.0 | 22.8 | BYOL | 18.3 |
| Prelax (ours) | **51.1** | Prelax (ours) | **38.1** | **23.3** | **23.9** | Prelax (ours) | **19.5** |
| | | Supervised | **39.4** | **24.2** | **25.3** | Supervised | **20.4** |

the standard Mask R-CNN [6] with FPN [8]. We compare the performances of models initialized with different pretrained weights on COCO:

- **RandInit**: randomly initialized weights;
- **BYOL**: unsupervised pretrained weights with BYOL;
- **Prelax** (ours): unsupervised pretrained weights with Prelax;
- **Supervised** (oracle): supervised pretrained weights.

From Table 5, we can see that even on the large-scale dataset, our Prelax still has a clear advantage over BYOL on all downstream tasks, including both in-domain linear evaluation and out-of-domain instance segmentation and object detection tasks.

## E  Sensitivity Analysis of Prelax Coefficients

Here we provide a detailed discussion on the effect of each coefficient of our Prelax objectives. We adopt the default hyper-parameters unless specified. For Prelax-std, it has three coefficients, the R2S interpolation coefficient $\alpha$, the similarity loss coefficient $\beta$, and the predictive loss coefficient $\gamma$. From Figure 4a, we can see that a positive $\alpha$ introduces certain degree of residual relaxation to the exact alignment and help improve the downstream performance. The best accuracy is achieved with a medium $\alpha$ at around 0.5. In addition, a large similarity coefficient $\beta$ tends to yield better performance, showing the necessity of the similarity constraint. Nevertheless, too large $\beta$ can also diminish the effect of residual relaxation and leads to slight performance drop. At last, a positive PL coefficient $\gamma$ is shown to yield better representations, although it might lead to representation collapse if it is too large, *e.g.,* $\gamma > 0.5$.

For Prelax-rot, as shown in Figure 4b, the behaviors of $\beta$ and $\gamma$ are basically consistent with Prelax-std. Nevertheless, we can see that only $\alpha = 1$ can yield better results than the SimSiam baseline, while other alternatives cannot. This could be due to the fact that the residual relaxation involves the first view $\mathbf{x}_1$ and its rotation-augmented view $\mathbf{x}_3$, and the R3S loss is designed between $\mathbf{x}_3$ and the second view $\mathbf{x}_2$. Therefore, in order to align $\mathbf{x}_3$ and $\mathbf{x}_2$ like the alignment between $\mathbf{x}_1$ and $\mathbf{x}_2$, all the relaxation information in $\mathbf{x}_3$ (which $\mathbf{x}_1$ does not have) must be accounted for, which corresponds to $\alpha = 1$ in R3S loss. We show that incorporating the rotation information in this way will indeed richer representation semantics and better performance.

Besides, we also find that in certain cases, adopting a reverse residual $\mathbf{r}_{21}$ in the R2S loss can bring slightly better results. In Figure 5, we investigate this phenomenon by comparing the normal and reverse residuals in R2S loss (applied for Prelax-std and Prelax-all) and R3S loss (applied for Prelax-rot). We can see that for R2S loss, using a reverse residual improves the accuracy by around 0.3 point, while for R3S loss, the reverse residual leads to dramatic degradation. This could be due to that R2S relaxes the gap between $\mathbf{x}_1$ and $\mathbf{x}_2$, whose representations are learned through swapped prediction in SimSiam's dual objective. Thus, we might also need to swap the direction of the residual to be consistent. Instead, in R3S, the relaxation is crafted between $\mathbf{x}_1$ and $\mathbf{x}_3$, so we do not need to swap the direction. Last but not least, we note that with the normal residual, Prelax-std and Prelax-all still achieve significantly better results than the SimSiam baseline, and the reverse residual can further improve on it.

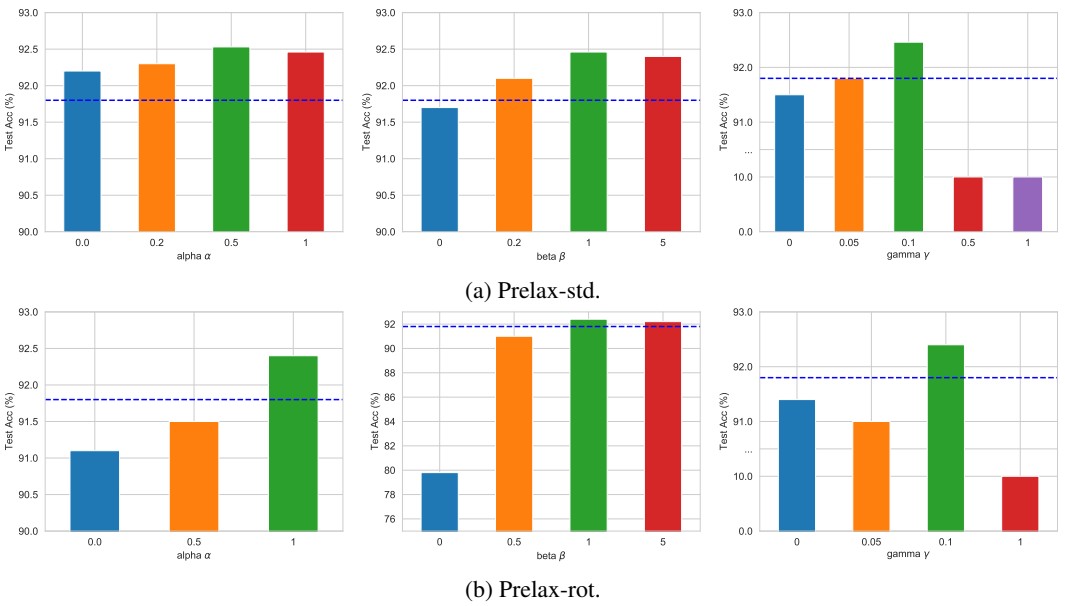

(a) Prelax-std.

(b) Prelax-rot.

Figure 4: Linear evaluation results of different Prelax-std and Prelax-rot coefficients on CIFAR-10 with SimSiam backbone. The dashed blue line denotes the result of the SimSiam baseline.

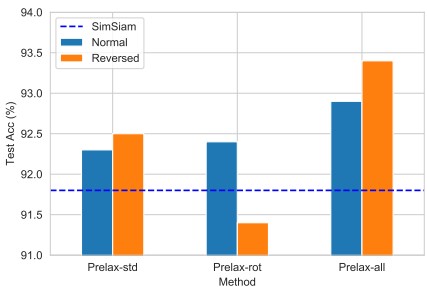

Figure 5: Comparison of normal and reverse residuals for Prelax variants on CIFAR-10 with Sim-Siam backbone.

## F  Learning Dynamics

In Figure 6, we compare SimSiam with Prelax-rot in terms of the learning dynamics. We can see that with our residual relaxation technique, both the relaxation loss and the similarity loss become larger than SimSiam. In particular, the size of the residual indeed converges to a large value with Prelax (1.1) than with SimSiam (0.6). As for the downstream classification accuracy, we notice that Prelax-rot starts with a lower accuracy, but converges to a large accuracy at last. This indicates that Prelax-rot learns to encode more image semantics, which may be harder to learn at first, but will finally outperform the baseline with better representation ability.