# OpenReview forum: "Residual Relaxation for Multi-view Representation Learning"
_NeurIPS.cc/2021/Conference — NeurIPS 2021 Poster_

### Official Review · Reviewer_wdr9 · 2021-07-12

**Rating:** 6
**Confidence:** 4

**Summary:**

The paper proposes a novel Pretext-aware Residual Relaxation (Prelax) approach to self-supervised learning that does not require exact alignment of the two different views, which can be harmful for views created from some types of augmentations, but introduces a residual vector that encodes the semantic shift caused by the augmentation. To ensure pretext-awareness, a loss is introduced to predict the pretext task from the residual vector. A theoretical analysis is provided to show Prelax's advantage over unsupervised approaches that either only consider multi-view learning or only predictive learning. Experiments indicate that adding Prelax to common self-supervised methods such as SimSiam and BYOL improves performance.

**Limitations And Societal Impact:**

The authors do not discuss the limitations of their proposed approach. While the reviewer agrees that the negative social impact should be limited due to its general methodology focus, there are certain limitations, such as the reliance on manually choosing the augmentations, which will be domain and dataset dependent.

**Main Review:**

Originality:
The proposed method bridges the gap between the contrastive/similarity-based methods, which learn pretext-invariant features, and the predictive learning frameworks that learn pretext-aware features. While previous work have attempted to do this in an ad-hoc manner by directly combining multiple losses or by using separate feature spaces, the authors propose a well-motivated approach of encoding the pretext-awareness into a residual alignment vector. Building on the theoretical framework developed in [1], they further provide an information theoretic perspective, illustrating the benefits of the proposed approach.

Quality:
The paper is technically sound and claims are well-motivated. However, an analysis of the interdependence and effects of the multiple losses would have been interesting. For instance by having a closer look at the gradients and monitor the alignment/mismatch of the various terms (or comparing them to the gradient of a supervised loss). Further, do the authors observe negligible improvements or performance degradation when increasing the number of views further beyond three?

Clarity:
The paper is overall easy to follow and the material is presented in a manner that should allow reproducibility.
- One statement, that could have been expanded a bit is the discussion of the results in the out-of-domain linear evaluation (2b). Do the authors have a hypothesis on why the Prelax-std/Prelax-rot perform slightly better than Prelax-all?
- The authors definition of multi-view seems a bit limited. Although multi-view representation learning includes the self-supervised techniques where additional views are created through augmentations, it commonly also includes settings where data is actually obtained from multiple views (different camera angles, or even different modalities).
- The claim that rotation causes severe accuracy drops is a bit strong and depends on the application domain. While it is a valid assumption for natural images, this is not the case in other domains. The authors might want to more explicitly include that this claim/work is addressing the case of natural images (although the method in itself is more general).

Significance:
The proposed approach is a relative generic framework that can be combined with most existing methods and should motivate additional work on trying to develop methods that exploit both pretext-invariant and pretext-aware features explicitly. Results illustrate a sufficient boost over the baseline approaches that only focus on similarity-based methods or predictive learning.

**Time Spent Reviewing:**

5

---

> ### Author Response · Authors · 2021-08-10
> **Response to Reviewer wdr9**
>
>
> Thanks for your detailed review and for appreciating the novelty and generality of our approach. We will address your concerns as follows.
>
> ---
> **Q1.** An analysis of the interdependence and effects of the multiple losses would have been interesting.
>
> **A1.** Yes, it is interesting to observe the learning dynamics of Prelax to study the roles of three objectives. In fact, in Figure 6 (Appendix E), we draw three plots to compare the learning behaviors between Prelax and SimSiam and we find:
> - **Relaxed alignment (2nd plot).** The similarity loss of Prelax is consistently larger than SimSiam, showing it indeed helps relax the exact alignment between $x_1$ and $x_2$.
> - **Larger semantic space for rotation (3rd plot).** To validate our rotation relaxation between $x_1$ and $x_3$, we further calculate the residual norm between $x_1$ and  $x_3$ (rotated $x_1$), which stands the encoded rotation semantics. We find that the residual norm of Prelax is also consistently larger than SimSiam, showing that Prelax-rot indeed has more freedom to encode the rotation semantics. Thus, our relaxation loss and predictive loss indeed help better encode the rotation semantics.
> - **Better test acc (1st plot).** As Prelax encodes both SimCLR as well as rotation augmentations, it solves a harder task than SimSiam. Therefore, its performance is relatively lower at the beginning stage but finally converges to a better representation quality than SimSiam.
>
> Overall, the learning dynamics verify our hypothesis and demonstrate the effectiveness of Prelax. We will elaborate more on this part in the revision.
>
> ---
> **Q2.** Do the authors observe negligible improvements or performance degradation when increasing the number of views further beyond three?
>
> **A2.** Currently, Prelax is built upon the existing contrastive learning framework between two augmented views $x_1,x_2$. In Prelax-rot, we intentionally craft a third view $x_3$ as a rotated version of $x_1$ to explicitly encode the rotation semantics. From this perspective, Prelax indeed paves a principled way to incorporate more views in contrastive learning by encoding higher-order similarities/differences into the residuals. We believe this is an interesting direction to explore, though it might be beyond the scope of this paper. We will continue to work on it.
>
> Besides, some existing strategies, like multi-crop [1], can also be directly integrated into Prelax to improve the performance by using more small-sized views. Likewise, we can also use residuals to account for their semantic differences.
>
> [1] Caron M, Misra I, Mairal J, Goyal P, Bojanowski P, Joulin A. Unsupervised learning of visual features by contrasting cluster assignments. In NeurIPS. 2020.
>
> ---
> **Q3.** Do the authors have a hypothesis on why the Prelax-std/Prelax-rot perform slightly better than Prelax-all in the out-of-domain linear evaluation (2b)?
>
> **A3.** Indeed, this is an interesting observation. First, we note that Prelax performs consistently better than SimSiam in both in-domain and out-of-domain linear evaluation. Also, we can notice there is inconsistency within Prelax variants. For example, Prelax-all performs better at in-domain evaluation while being (slightly) worse at out-of-domain evaluation.
>
> This phenomenon could be due to that Prelax-all encodes more image semantics of the training domain. Because it includes both Prelax-std and Prelax-rot objectives to encode as much information as possible, it is more biased towards the training distribution. Thus, when the distribution shifts across domains, its representation ability is more severely damaged compared to Prelax-std and Prelax-rot, resulting in a worse out-of-domain generalization ability.
>
> ---
> **Q4.** The authors' definition of multi-view seems a bit limited.
>
> **A4.** Thanks for pointing it out. We agree with you that multi-view learning covers many topics. For examples, views could come from different modalities, e.g., image+text, audio+video; or we can construct synthetic views, like word + context word, etc. While in this paper, we mainly focus on the Self-Supervised Learning (SSL) setting where we craft multiple views of natural images by random data augmentation. We will clarify it in the revision.
>
> ---
> **Q5.** The claim that rotation causes severe accuracy drops is a bit strong and depends on the application domain.
>
> **A5.**  In this paper, as we focus on natural images, we take rotation as a motivating example as it is harmful for natural images. Indeed, as you point out, rotation is not necessarily harmful in other domains, e.g. medical images.
> In fact, Prelax is a generic method and can incorporate other prior knowledge in specific domains. For medical images, random cropping could instead be very harmful for the exact alignment as the important part could lie in the corner of the image. In this case, our residual relaxation mechanism could also be used to encode the semantic shift caused by cropping and alleviate its bad effects.
>
> We will make this point more clear in the revision following your advice.
>
> ---
> Thank you again for perusing this work with fruitful comments. Hope you find our explanations helpful!

---

### Official Review · Reviewer_6qif · 2021-07-14

**Rating:** 6
**Confidence:** 3

**Summary:**

In this work, the authors propose a new approach, named Pretext-aware Residual Relaxation (Prelex) to learning the representations from multiple views. The method mainly employs the residual relaxed similarity loss to improve the alignment between positive samples.

**Limitations And Societal Impact:**

Yes

**Main Review:**

Majors.
1 Motivation. Fig. 1 (a) shows the main motivation that rotation augmentation causes severe accuracy drop in a standard supervised model. In line 34, the authors claim the rotation augmentation carries meaningful semantics while being harmful to existing multi-view methods due to a large semantic shift. Thus, they propose a generic approach to address this dilemma. I have two questions. (1) I am curious whether the setting in Fig. 1 (a) is right or suitable to conclude the rotation is harmful. The setting in Fig. 1 (a) is that they train the model using two data augmentations (random crop and random horizontal flip in line 69 of the supple), then evaluate the model on images under four data augmentations, including horizontal flip, grayscale, color jitter and rotation. The drop of accuracy on rotated images seems reasonable since the trained model doesn’t ‘see’ such data? Maybe, these experimental results cannot support their motivation. (2) If the setting in Fig.1 (a) is right, the authors should try to indicate the effectiveness of their proposed method to deal with the dilemma on the same setting in Fig. 1 (a). I don’t find any results about it.

2 Fig. 2 shows the network. But it needs more correlation with the equations in the manuscript and indications, the meaning of the symbols.

3 Comparison with existing work [5] and [10]. The authors need to provide more comparisons to list the similarities and differences in a table format, including the key loss items, such as the residual relaxed similarity loss, predictive learning loss, and constraint on the similarity. The comparison is used to highlight the novel part of Prelax. After reading the whole manuscript, at the first time, I am confused that the proposed Prelax is an individual loss combination (in Section 3) or some additional parts used in SOTA [5][10] (such as the items, ‘SimSiam + Prelax’ or ‘BYOL+Prelax’ in Table 1)

4 In section 4, they design three variants of Prelax. I want to know whether they investigate the dual order in Prelax-std, rot, all. For example, in eq (14), the order is $r_{12}$. How about the $r_{21}$. Also for eq (15,15,17,18,19,20).

5 Experiment. The datasets used in this work include CIFAR-10, CIFAR-100, Tiny-ImageNet-200 and ImageNette, which might not support a wide range of benchmark datasets. The authors need to report the results on more downstream tasks as the table 5 in [5], such as detection and segmentation.


**Time Spent Reviewing:**

3 hrs

---

> ### Author Response · Authors · 2021-08-10
> **Response to Reviewer 6qif**
>
> Thanks for your detailed and insightful review. We will address your concerns as follows.
>
> ---
> **Q1.** Motivation. (1) Is the setting in Fig. 1 (a) right or suitable to conclude the rotation is harmful? (2)  If the setting in Fig.1 (a) is right, the authors should try to indicate the effectiveness of their proposed method to deal with the dilemma on the same setting in Fig. 1 (a).
>
> **A1.** Here, we try to answer your questions about our motivation part.
>
> a) Harmfulness of Rotation
>
>  In the experiment of Figure 1, we follow a standard supervised training protocol with the two basic augmentations. More rigorously, we **remove all data augmentations** in the training stage, and re-evaluate the five augmentations at the test stage.
>
> | Test Transformation| Acc (%) |
> | ---| --- |
> | None (baseline) | 89.7 |
> | RandomHorizontalFlip | 89.6 |
> | ColorJitter | 78.9 |
> | RandomGrayscale | 88.7 |
> | RandomRotation | **50.9** |
>
>
>
> We can see that the results are still similar to those in Figure 1, and the neural network is still much more vulnerable to rotation than the rest. Thus we can conclude that rotation is indeed a more harmful augmentation.
>
> **Further evidence in unsupervised learning.** To further show how rotation is harmful for unsupervised learning, we also report the results of adding rotation to SimSiam's training process. As reported in Table 3(a) ("SimSiam + rotation aug."), the performance drops from 91.8 to 88.3, showing that the rotation augmentation is indeed harmful for unsupervised methods. Besides, the harmfulness of rotation is also observed in other methods. For example, Figure 5 in the SimCLR paper [1].
>
> We will add this discussion in this revision.
>
> [1] Chen T, Kornblith S, Norouzi M, Hinton G. A simple framework for contrastive learning of visual representations. In ICML. 2020.
>
> b) How does our method address this dilemma?
>
> Figure 1 is used to illustrate the harmfulness of rotation and we take it as a motivating observation for our method, and the same phenomenon can also be observed in unsupervised learning (see discussions above).
>
> As our method focuses on solving this problem for multi-view learning with residual relaxation, it cannot be directly adapted to supervised learning as in Figure 1. Nevertheless, we still demonstrate its effectiveness for unsupervised learning.
>
> Specifically, in Table 3(a), we have compared our Prelax with several other options to incorporate the rotation augmentation to SimSiam, including both 1) adding to the augmentation list, 2) directly combining two objectives (SimSiam + Rotation), and 3) margin loss, and find these options can either be harmful or bring little improvements. In comparison, Prelax can significantly benefit from the rotation augmentation by improving from 91.8 to 93.4.
>
> This shows that our proposed residual relaxation mechanism can resolve the dilemma with a strong but meaningful augmentation.
>
> We will make this discussion more clear in the revision.
>
> ---
> **Q2.** Fig. 2 needs more correlation with the equations in the manuscript and indications.
>
> **A2.** Thanks for your suggestion and we will add more explanations to the diagram.
>
> ---
> **Q3.** Comparison with existing work, SimSiam, and BYOL, to highlight the novelty part by Prelax.
>
> **A3.** Following your suggestion, we explicitly compare the similarities and differences between SimSiam / BYOL (the two have the same objectives, cf. Section 4.1) and Prelax objectives in the following table.
>
> |Objective | Relaxation | Similarity  | Predictive Learning |
> | --- | --- | --- | --- |
> | SimSiam / BYOL | sim loss ($x_1,x_2$) | sim loss $(x_2,x_1$) | - |
> | Prelax-std | R2S loss ($x_1,x_2$) | sim loss $(x_2,x_1$) | PL loss ($r_{12}$) |
> | Prelax-rot | R3S loss ($x_1,x_2,x_3$) | sim loss $(x_2,x_1$)| RotPL loss ($r_{13}$) |
> | Prelax-all | R2S loss ($x_1,x_2$) + R3S loss ($x_1,x_2,x_3$) | sim loss $(x_2,x_1$)| PL loss ($r_{12}$) + RotPL loss ($r_{13}$) |
>
> In Section 3, we present the general methodology of Prelax in a unified framework of multi-view learning. In Section 4, we show how to implement it on two SOTA backbones, SimSiam and BYOL, which results in the two practical variants of Prelax. Our Prelax variants **only modify the objectives (as shown above) of the corresponding backbone, and leave all the rest unchanged**. We have a detailed description of the two variants in Section 5 (Lines 216-225) and Appendix B.
>
> We will make it more clear in the revision.
>
> ---
> **Q4.** Do they investigate the dual order in Prelax-std, rot, all?  For example, in eq (14), the order is $r_{12}$. How about the $r_{21}$. Also for eq (15,15,17,18,19,20).
>
> **A4.** In fact, $x_1$ and $x_2$ are typically randomly sampled from the same augmentation list (i.e., identically distributed), so there is no difference between using $r_{12}$ or $r_{21}$. The two are equivalent in expectation.
>
> We will add this discussion in the revision.
>
> ---
> **Q5.** Report the results on more downstream tasks, such as detection and segmentation.
>
> **A5.** In the paper, we have conducted experiments on a range of benchmark datasets, from CIFAR-10, CIFAR-100, to Tiny-ImageNet-100, and evaluate both in-domain and out-of-domain generalization. Following your suggestions, we conduct additional experiments as follows.
>
> **Setup.** Although we cannot carry out the full ImageNet experiments with limited time and computation, we gather some preliminary results on the downsampled ImageNet dataset  (128x128) with the ResNet-18 backbone. For a fair comparison, our experiments are conducted with the official code of BYOL. All models are trained for 100 epochs with the default hyperparameters.
>
> **Evaluation protocol.** For downstream evaluation, we report both 1) linear evaluation for classification on ImageNet, and 2) transfer learning for the object detection task on COCO. We conduct the object detection experiments on the standard RetinaNet with FPN, and compare the performances of models initialized with different pretrained weights on COCO:
> - RandInit: randomly initialized weights;
> - BYOL: unsupervised pretrained weights with BYOL;
> - Prelax: unsupervised pretrained weights with Prelax;
> - Supervised: supervised pretrained weights.
>
> a) Linear Evaluation
>
> |Method | TOP-1 (%) |
> | --- | --- |
> | BYOL | 49.2 |
> | Prelax | **51.1** |
>
> b) COCO Object Detection
>
> | Pretraining Methods | AP$_{50}$ | AP | AP$_{75}$ |
> | --- | --- |  --- | --- |
> | RandInit | 32.7 | 19.5 | 20.1 |
> | BYOL | 36.6 | 22.0 | 22.8 |
> | Prelax | **38.1** | **23.3** | **23.9** |
> | Supervised | **39.4**  | **24.2** | **25.3** |
>
> We can see that even on the large-scale dataset, our Prelax still has a clear advantage over BYOL on both linear evaluation and object detection tasks.
>
> ---
>
> Thank you again for reviewing our paper, and hope our explanations are helpful. We are willing to take your further questions before the discussion session ends.

---

> > ### Comment · Reviewer_6qif · 2021-08-16
> > **Response**
> >
> > Thanks very much for your detailed responses. I have carefully read the feedbacks and other reviews. I decide to upgrade my rating (above the acceptance threshold). The authors have done a wonderful job and addressed my most questions, such as the important motivation, the comparisons, and downstream work. I have no further questions.

---

> > > ### Author Response · Authors · 2021-08-17
> > > **Thanks and a kind reminder**
> > >
> > > Thanks a lot for your careful reading and for appreciating our response. We are very glad that you find our explanations helpful. We also notice that although you mentioned that you will upgrade your rating above the acceptance threshold, the rating in your original review has not been updated yet. This is just a kind reminder as you might be too busy and forget to update it. Thank you again for appreciating our work.

---

### Official Review · Reviewer_TbTy · 2021-07-17

**Rating:** 7
**Confidence:** 4

**Summary:**

The paper focus on the representation learning using a relaxed alignment based on the fact that some strong data augmentations may hurt the performance. The authors introduced an adaptive residual vector and designed several novel objectives to make the residual vector useful and meaningful. Experiments shows a significant improvement over various backbones.

**Limitations And Societal Impact:**

No significant social risk

**Main Review:**

Strengths:
- The paper is well organized and easy to follow. The proposed prelax method is well motivated and general. Prelax can be used for various data augmentations even there is some semantic shift, which is very important for real-world applications.
- Prelax is also a unified approach to bridge the pretext-invariant features and pretext-aware features for representation learning.
- The authors also provided the theoretical analysis based on the information-theoretic framework, which shows that Prelax provably enjoys better downstream performance.

Weakness:
- Prelax has three versions, Prelax-std, Prelax-rot and Prelax-all, then which one should be used when facing with an application?
- How to choose the hyperparameters alpha and gama?
- The difference between R2S and R3S is not clear.

**Time Spent Reviewing:**

5

---

> ### Author Response · Authors · 2021-08-10
> **Response to Reviewer TbTy**
>
>
> Thank you for your valuable feedback. We address your concerns as follows.
>
> ---
> **Q1.** Prelax has three versions, Prelax-std, Prelax-rot, and Prelax-all, then which one should be used when facing an application?
>
> **A1.** We design the three practical versions as different implementations of our generic framework of residual relaxation. Among them, Prelax-std focuses on further cultivating existing augmentation strategies, Prelax-rot is to incorporate the stronger (potentially harmful) rotation augmentation, while Prelax-all combines them all. Through the three versions, we demonstrate the wide applicability of Prelax.
>
> As for practical users, they could also adapt Prelax to their own application by incorporating specific domain knowledge. Here are our suggestions in terms of the three variants:
> - **Prelax-std: plug-and-play**. If they already have a mature pipeline for contrastive learning with effective augmentations, like SimCLR and SimSiam, then Prelax-std can boost its performance with minimal change to the existing methods and minimal computation overhead.
> - **Prelax-rot: strong-but-meaningful augmentations**. If they happen to know there is a semantically meaningful augmentation for the application but it is harmful for contrastive learning, like image rotation for natural images, and random cropping for medical images, then, Prelax-rot is an exemplar for such scenarios.
> - **Prelax-all: best-of-both-worlds.** If they happen to have both effective contrastive-learning augmentations (like SimCLR ones) as well as a strong but meaningful one (like rotation), then Prelax-all can encode both kinds of augmentations and achieve the best performance.
>
> We will incorporate the above discussions in the revision.
>
> ---
> **Q2**: How to choose the hyperparameters alpha and gamma?
>
> **A2**: We find that our algorithms are relatively robust to hyperparameters. The three Prelax variants all work well under **the same vanilla setting** ($\alpha=1,\beta=1,\gamma=0.1$). Here $\alpha=1$ corresponds to the full residual relaxation (Eq. 2), $\beta=1$ means the two alignment terms are balanced, and $\gamma=0.1$ is chosen as a small regularization.
>
> More rigorously, we also conduct a sensitivity analysis of the three hyperparameters in **Appendix D**, where we can see that the default parameters indeed perform well enough.
>
> ---
> **Q3.** The difference between R2S and R3S is not clear.
>
> **A3.** R2S loss is designed to encode the semantic shift between the canonical augmentations applied to $x_1$ and $x_2$, like that in SimCLR and SimSiam. So in R2S the residual term is $r_{21}=z_2-z_1$.
>
> Instead, R3S introduces a third view $x_3$ as a randomly rotated view of $x_1$, and now the residual *merely* stands for the semantic shift brought by the rotation, so we have $r_{31}=z_3-z_1$. Replacing $z_1$ by $z_3 - r_{31}$ in the similarity loss between $x_1$ and $x_2$, we will have the R3S loss.
>
> In conclusion, the relaxation of the R2S loss accounts for all the semantic shift between $x_1$ and $x_2$, while that of the R3S loss only accounts for the rotation augmentation between $x_1$ and $x_3$. We will elaborate on this part better in the revision.
>
> ---
> Thank you again for your review, and hope you find our explanations helpful!

---

### Official Review · Reviewer_hQVQ · 2021-07-24

**Rating:** 6
**Confidence:** 3

**Summary:**

The paper proposes a residual relaxation method to encode the semantic shift for the multi-view self-supervised learning. The method relaxes the exact alignment for the latent vector representation from different views. The experimental results show the method can improve the performance of the data augmentation.

**Ethical Concerns:**

No.

**Limitations And Societal Impact:**

Yes.

**Main Review:**

1. Originality: The work is a novel combination of the multi-view representation learning and the predictive learning, and proposes residual relaxation to improve the performance.

2. Quality: The work is technically sound. But, from the ablation study, the predictive learning term contributes more than the proposed residual relaxation technique.

3. Clarity: The paper is well written and organized.

4. Significance: The result has referential significance. The other researchers could build on the idea of residual relaxation for other applications.

**Time Spent Reviewing:**

12

---

> ### Author Response · Authors · 2021-08-10
> **Response to Reviewer hQVQ**
>
>
> Thank you for your valuable review. We address your concerns as follows.
>
> ---
> **Q1**. From the ablation study (Table 3b), the predictive learning (PL) term contributes more than the proposed residual relaxation technique.
>
> **A1.** The ablation study in Table 3b is designed to illustrate that all of the three terms in Prelax-std and Prelax-rot are necessary. In particular, we can see that removing the PL loss leads to a large performance drop, showing that it is an important part of Prelax. Nevertheless, we want to highlight that **PL loss alone cannot improve over SimSiam consistently**, as shown below.
>
> a) Prelax-std: existing augmentations
>
> |Method | Acc. (%)|
> |------ |----|
> |Sim (SimSiam) | 91.8 |
> |Sim + PL | 92.2 |
> |Sim + PL + R2S (Prelax-std) | **92.5** |
>
> b) Prelax-rot: incorporating the rotation augmentation
>
> |Method | Acc. (%)|
> |------ |----|
> |Sim (SimSiam) | 91.8 |
> |Sim + RotPL | 91.1 |
> |Sim + RotPL + R3S (Prelax-rot) | **92.4** |
>
> From the tables above, we can see that:
> - A naive combination of SimSiam and PL losses cannot always improve over the SimSiam baseline. For example, in Table b, it leads to 0.7 drop in accuracy when facing a strong augmentation (rotation).
> - Incorporating our residual relaxation with the PL loss, we can improve over the baselines significantly and consistently. In Table a, our R2S loss further improves 0.3 with existing augmentations (Prelax-std). More importantly, in Table b, the R3S loss improves from 91.1 to 92.4, and successfully alleviates the bad effect of PL loss.
>
> Thus, we can see that our residual relaxation (R2S and R3S) is the key to improve over SimSiam significantly and consistently, while it has to work together with PL loss to guarantee that the residual is pretext-aware. In other words, **PL is necessary, but residual relaxation is the driving force**.
>
> We will add this discussion in the revision.
>
> ---
>
> Please let us know if you have any additional questions or require further clarifications. We are happy to address them before the discussion session ends.

---

### Decision · Program_Chairs · 2021-09-27

**Decision:**

Accept (Poster)

**Comment:**

This paper relaxes the alignment objective in multiview self-supervised learning when the data augmentation causes semantic shifts in different views. The proposed pretext-aware residual relaxation method allows an adaptive residual vector between different views. The proposed method can benefit from stronger image augmentations like rotation and outperform the existing methods. Based on a recent theoretical framework on self-supervised learning, the authors provide theoretical guarantees of the proposed method. The paper is well motivated, well written and has practical impact on downstream image classification tasks. Reviewers all agreed that the proposed solution is interesting and the improvement is significant. I am recommending acceptance of this paper.  The authors need to make sure they add the new results in the rebuttal to the final version.